# Comparing acoustic and radar deterrence methods as mitigation measures to reduce human-bat impacts and conservation conflicts

**Lia R. V. Gilmour** [1]*, **Marc W. Holderied**[1], **Simon P. C. Pickering**[2], **Gareth Jones**[1]

**1** School of Biological Sciences, University of Bristol, Bristol, United Kingdom, **2** Ecotricity Group Limited, Stroud, Gloucestershire, United Kingdom

* Lia.Gilmour@bristol.ac.uk

## Abstract

Where humans and wildlife co-exist, mitigation is often needed to alleviate potential conflicts and impacts. Deterrence methods can be used to reduce impacts of human structures or activities on wildlife, or to resolve conservation conflicts in areas where animals may be regarded as a nuisance or pose a health hazard. Here we test two methods (acoustic and radar) that have shown potential for deterring bats away from areas where they forage and/ or roost. Using both infrared video and acoustic methods for counting bat passes, we show that ultrasonic speakers were effective as bat deterrents at foraging sites, but radar was not. Ultrasonic deterrents decreased overall bat activity (filmed on infrared cameras) by ~80% when deployed alone and in combination with radar. However, radar alone had no effect on bat activity when video or acoustic data were analysed using generalised linear mixed effect models. Feeding buzzes of all species were reduced by 79% and 69% in the ultrasound only treatment when compared to the control and radar treatments, but only the ultrasound treatment was significant in post-hoc tests. Species responded differently to the ultrasound treatments and we recorded a deterrent effect on both *Pipistrellus pipistrellus* (~40–80% reduction in activity) and *P. pygmaeus* (~30–60% reduction), but not on *Myotis* species. However, only the ultrasound and radar treatment was significant (when compared to control and radar) in post-hoc tests for *P. pipistrellus*. Deterrent treatment was marginally non-significant for *P. pygmaeus*, but the ultrasound only treatment was significant when compared to radar in post-hoc tests. We therefore suggest that acoustic, but not radar methods are explored further as deterrents for bats. The use of acoustic deterrence should always be assessed on a case-by-case basis, with a focus on bat conservation.

## Introduction

With an ever-expanding world population, increased incidences of human-wildlife interactions are inevitable. These interactions can lead to detrimental impacts on wildlife, ranging from habitat loss to direct mortality and in some cases can also have significant impacts on human lives [1]. The phrase 'human-wildlife conflict' can be misleading, as it pitches humans

**Funding:** This study was funded by the National Environment Research Council (NERC) (NE/K007610/1), with CASE contribution from Ecotricity (www.ecotricity.co.uk), received by LRVG and PI was GJ. The funders (NERC) had no role in study design, data collection and analysis, decision to publish, or preparation of the manuscript. SPCP of Ecotricity Group Limited also provided a supervisory role in the project and review of the manuscript before submission for publication.

**Competing interests:** This study was part of a PhD funded by the National Environment Research Council (NERC) (NE/K007610/1). As part of the PhD funding, £3000 was provided towards stipend costs for LRVG as a CASE contribution by the commercial funder Ecotricity Group Limited (www.ecotricity.co.uk). SPCP of Ecotricity Group Limited also provided a supervisory role in the project and review of the manuscript before submission for publication. This does not alter our adherence to PLOS ONE policies on sharing data and materials.

and animals against each other [2]. In areas where humans and wildlife coexist, conflict often arises between stakeholders wishing to conserve species and those who have other more anthropocentric interests [2]. Therefore, the phrases 'human-wildlife impacts' and 'conservation conflicts' more adequately sum up human-wildlife interactions [2, 3].

Human-wildlife impacts and conservation conflicts are often key conservation concerns for a wide range of taxa [3]. Many human structures and activities put species at risk, for example by inducing changes in behaviour, degrading habitats or by producing a direct mortality risk [4–6]. Conservation research has therefore focussed on mitigation strategies that exclude animals from areas where they may come to harm, or where they may impact or cause damage to human activities/structures, for example using deterrence [7–15].

Bats are one group for which deterrence has been suggested as a potential mitigation method for reducing human impacts and conservation conflicts [12, 16–19]. Protecting bat species is a key conservation concern for many European countries and many species have legal protection and/or are listed as 'Endangered' [20]. Being slow to reproduce, long-lived and subject to the high energy requirements of flight, bats are regarded as vulnerable to threats such as habitat loss, climate change and emerging diseases [20]. Human structures and activities can also put bat populations at risk and are therefore present potential applications for deterrence. For example, wind energy development and construction pose significant threats to bat populations, with large numbers of fatalities documented in North America in recent decades [21, 22]. Fatalities in Europe are not documented on such a large scale, but still have the potential to cause deleterious effects to resident and migratory bat populations [23–27]. Deterrence may therefore provide a way to keep bats away from the rotor-swept zone of wind turbines, reducing fatalities [16]. Similarly, roads and other transport infrastructures are also likely to have significant negative impacts on bat populations, due to habitat loss, noise pollution and mortality due to collision, yet mitigation is at present mostly insufficient [28]. Deterrence could be used alongside existing mitigation such as green bridges or overpasses, diverting bats away from flight lines over roads where they may be at risk from collision, towards safer routes.

Bats can also cause damage and a cleaning burden in historic buildings where they roost [12] and in some cases pose a human health hazard in workplaces, schools and places of worship [29, 30]. Conservation conflicts can therefore arise between those wishing to conserve bat populations and people using a building where bats are roosting and causing problems (usually with their urine and faeces) [12, 29]. In some cases, this leads to exclusion of bats from buildings, which if carried out unlawfully or without careful consideration, can be detrimental to a colony's survival [30].

Therefore, reducing the impact of human structures, activities and conservation conflicts on bat populations, is crucial for bat conservation, and should be a priority. However, before a deterrence method is implemented, exploration of alternatives should be undertaken, following the mitigation hierarchy, first seeking to avoid or minimise any impacts before moving to reduce or compensate [31]. Indeed, using deterrence to move bats out of an area may cause unintended effects, such as habitat loss, barrier effects and/or stress, which need to be weighed up against alternative mitigation [32]. However, where more benign alternatives fail, deterrence is a method that should be considered, especially in situations where bats are at risk of serious harm or direct mortality.

Potential deterrence methods for bats include light, radar and sound [12, 16, 18, 33–35]. Some bat species are deterred by certain types of lighting, for example streetlights or flood-lights [33, 36–38]. Lighting has recently been tested as a potential bat deterrent for use in churches where urine and faeces can cause damage and a cleaning burden [12, 39]. Illumination of 'no-fly zones' within churches limited use of those areas, but bats became entombed in roosts that

were directly lit, causing the authors to caution against using this method without careful consideration and further investigation [12]. Indeed, lighting in buildings where bats roost can be problematic and even detrimental for species that roost there, causing delays in emergence, roost abandonment, habitat fragmentation, effects on commuting, foraging and hibernation, and in some cases death [33, 40]. One striking example was the report of a 40watt light bulb causing the sudden deaths of over 1000 *Myotis myotis* bats in Germany, when it was left on inside a roost for two days [40]. Roosting colonies of *Plecotus* species were also significantly reduced in churches in Sweden, where flood-lights had been installed, when compared to non-lit churches over the same 25+ years [38]. Ultraviolet (UV) light has also shown potential as a deterrent for Hawaiian hoary bats (*Lasiurus cinereus semotus*) at wind turbines, where they are at risk of collision [35]. However, light has the potential to be attractive to some bat species, rather than a deterrent [33, 37], probably due to its attractive effect on insects, which often aggregate around street lights, especially those that emit UV [41–43]. Due to the potential for adverse effects of light deterrents for bats outweighing the potential benefits, we decided not to include these methods in this study and focussed instead on radar and acoustic deterrents.

Radar has been proposed as a bat deterrent, specifically for use at wind turbines [18, 19]. Suggested mechanisms of deterrence include a thermal burden effect of the electromagnetic radiation, or a jamming effect on echolocation, by high frequency sound produced during thermoelastic expansion of brain tissue [18]. Bat activity and foraging effort were lower at air traffic control regions in Scotland where radar was deployed [18]. A follow-on study, also in Scotland, recorded reduced bat echolocation call activity and foraging when an X-band (8–12 GHz) marine radar unit was used at riparian foraging sites [18].

The potential for using ultrasound (high frequency sound above 20 kHz) to deter bats has also received attention [12, 16]. Foraging bats tend to avoid noise, whether from natural sources (e.g. high frequency sounds emitted by insects and/or produced by turbulent water) [44, 45] or anthropogenic noise pollution (e.g. traffic noise) [46–49]. Ambient noise may deter bats, as it precludes the use of echolocation for prey detection or orientation, masks sounds made by insect prey, or simply because it produces a disturbing airspace [44–48]. Sources of ultrasound may also affect a bat's ability to communicate with conspecifics or indeed the ability to eavesdrop on echolocation calls of other bats (which can be beneficial for finding foraging sources and mates) [50, 51]. Bats are also affected by high frequency ultrasonic clicks produced by some noxious moths (and their mimics) and this is due, in part to a 'jamming' mechanism that leaves the bat's echolocation system unusable [52–55]. Therefore, bat deterrent systems are usually designed with the aim to mask or jam the echolocation calls of bats [16]. Field trials of acoustic deterrents at wind farm sites in North America were successful in reducing the numbers of bat fatalities [16]. The same speakers also showed potential in reducing conservation conflicts in historic buildings, where bat droppings and urine can potentially damage valuable historic artefacts [12, 39].

Deterrence is therefore a method that has potential in cases where other measures have failed, to reduce impacts of human structures and activities on bats and alleviate conservation conflicts in areas where bats roost and come into contact with humans. Our study compares the effectiveness of radar and acoustic bat deterrents at foraging sites in the UK. We also discuss the potential for these deterrents to be used in mitigation more widely.

## Materials and methods

### Site selection, equipment and experimental procedure

We carried out experiments during June-September 2015 at 16 riparian sites (>1 km apart, to minimise the chances of recording the same bats) that contained a stretch of river or canal with an area of still water and a bridge, chosen as they were likely to have relatively high

concentrations of foraging bats [56]. River sites were located on the border of England and Wales in Herefordshire/Shropshire and Powys, (area range 52˚25'16.62"N, 3˚1'34.14"W to 52˚ 20'55.92"N, 2˚59'0.59"W) and canal sites in Gloucestershire and Somerset (51˚49'22.81"N, 2˚ 17'55.97" to 51˚22'57.54"N, 2˚23'50.84"W). We deployed ultrasonic speakers (Deaton engineering Inc., Texas, USA) and radar (X-band Marine Radar FR-8062, Furuno Electric Co., Ltd, Tokyo, Japan), together and in isolation, alternately with a silent control (no sound/radar) for 10 minutes per treatment, over 4 treatment time blocks (with a 5-minute recovery period after each treatment time block where no sound/radar was deployed). Experiments lasted 1 hour, starting 30–45 minutes after sunset (depending on ambient light levels), when bat activity was likely to be at its highest. We alternated the four treatments over the 16 sites, following a temporal Latin square design, so that each of the four treatments was deployed at least once in each of the four 10-minute time blocks. However, we could not use data from two sites in the analysis due to equipment failure. We therefore performed statistical analysis on the data using generalized linear mixed effect models (GLMMs), which control for unbalanced order in experimental design [57]. We also included time block as a fixed effect in statistical analysis, to control for temporal changes in bat activity during the experiment.

We used a near-infrared (NIR) security camera (Y-cam HD, Y-cam Inc., Twickenham, UK), with IR illumination (2 LED lamps, ShantouScene, Shenzhen, China) and a laptop computer to film the 'treated airspace' for 1 hour (Fig 1). We recorded bat activity acoustically using an SM2BAT+ bat detector with an SMX-US omnidirectional ultrasonic microphone (Wildlife Acoustics Inc., Massachusetts, USA; continuous .wav recording; 384 kHz sampling rate; SNR 10), placed at the edge of the treatment zone at ~20 m from the NIR camera (covering a range of ~30 m). We placed the radar unit on a table at 0.5 m in front of the camera, with the antenna at 1 m height, in the fixed position (rather than rotating) and set it to emit a pulse length of 0.3 μs (repetition rate 1200 Hz, peak power 6 kW), as this was the most likely set-up and duty cycle to affect bat activity [18] (beam width: horizontal = 1.9˚, vertical = 22.0˚). We placed ultrasonic speaker units on chairs at ~0.5 m height and ~0.5 m apart and angled them in the same direction as the radar beam towards the middle of the treatment area, perpendicular to the baseline of the treatment area (where the camera was placed). Speakers were the same units as used at a wind energy facility in North America [16] and in churches in the UK [12, 39] and had 16 transducers capable of emitting continuous broadband ultrasound at 20–100 kHz, with a frequency of most energy of 50 kHz (Senscomp, Michigan, USA; source level at 1 m 110 dB SPL min at 50 kHz; 85 dB SPL at 20 meters, 20˚C, 10% relative humidity and 101.33 kPa [16]; beam angle 15˚ at -6 dB) (Fig 2). The speakers were chosen for testing as their frequency of most energy overlapped with the echolocation call frequencies of British bat species likely to be present during experiments, including *Pipistrellus pipistrellus* (frequency of most energy 45 kHz), *P. pygmaeus* (55 kHz) and *Myotis* species (30–50 kHz) (Fig 2). We recorded temperature every 15 minutes using a Watson W-8681-SOLAR weather station (Watson Inc., Beijing, China). Experiments were carried out on low wind nights (wind speeds <5 m/s) when there was no rainfall forecast, as wind and rain can reduce bat activity. We powered the camera and acoustic deterrents using a low-noise generator (Honda EU10i, Honda, Tokyo, Japan), placed >10 m from the treatment area, and the radar unit by a Bosch 12 V battery (Robert Bosch Ltd., Uxbridge, UK). The generator was running during both control and treatment periods and a previous study using a similar model found no generator effect on bat activity [34].

## Video and acoustic analysis

We calculated bat activity from NIR footage for each treatment (number of bats moving in and out of frame) using Quick Time Player (v7.7.8, Apple Inc, Cupertino, USA). Video

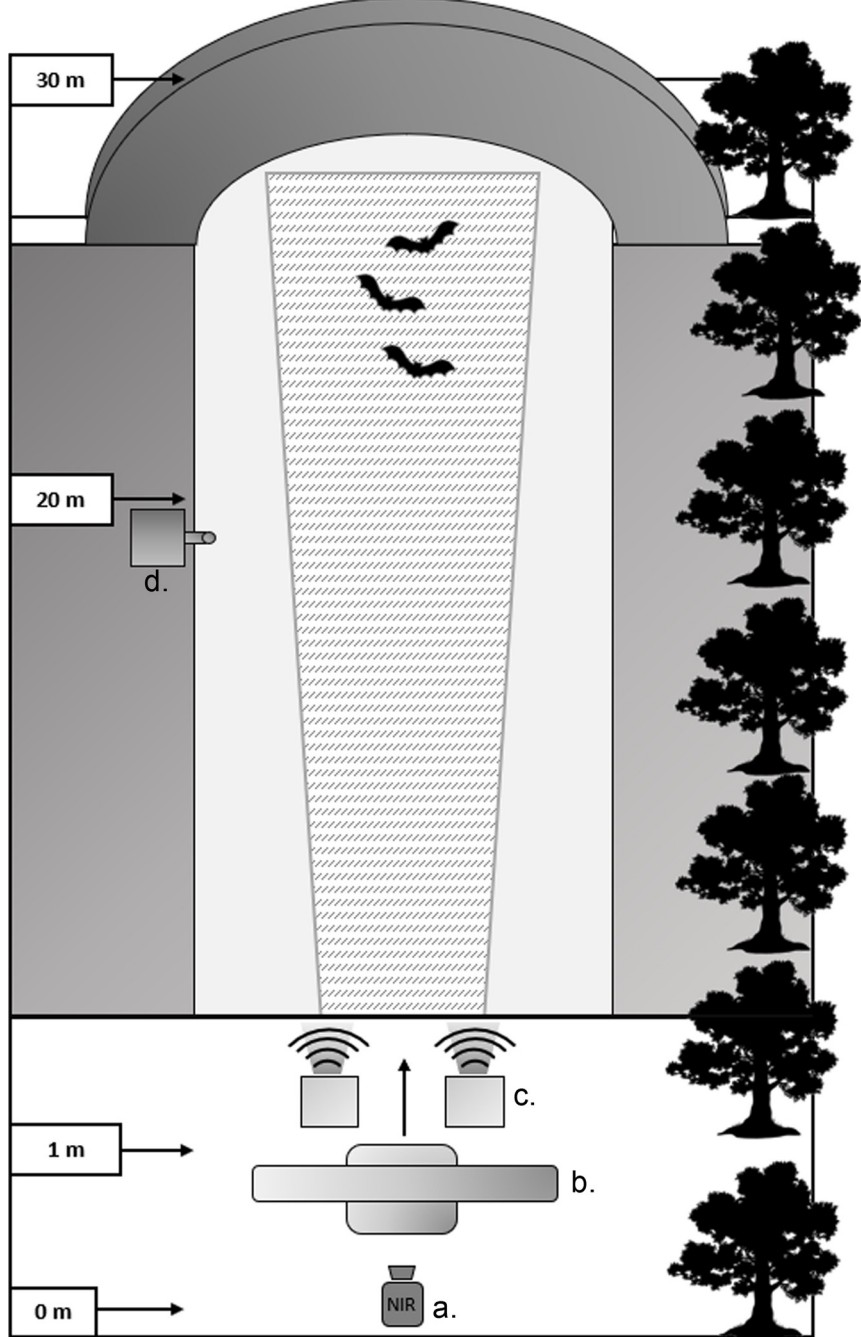

**Fig 1. Schematic of example experiment set-up at riparian sites.** Schematic of experiment set-up at riparian sites, including treated airspace (textured area) in an area of river/canal with a bridge, flanked by a tree line or hedge on one side. Near infrared (NIR) video camera and laptop were set up at 0 m (a.), the radar unit on a table at 0.5 m from the camera (b.), the acoustic deterrent speaker units on chairs at 1 m (c.) and the SM2 BAT+ bat detector at ~20 m (d.).

analysis was carried out blind to treatment and results were validated by an independent observer (for five randomly selected sites). We identified bat passes manually to genus or species level where possible using Bat Sound (v4.1.4, Pettersson, Uppsala, Sweden; FFT size: 1024; FFT window: Hanning) and calculated acoustic bat activity, as the number of passes or feeding

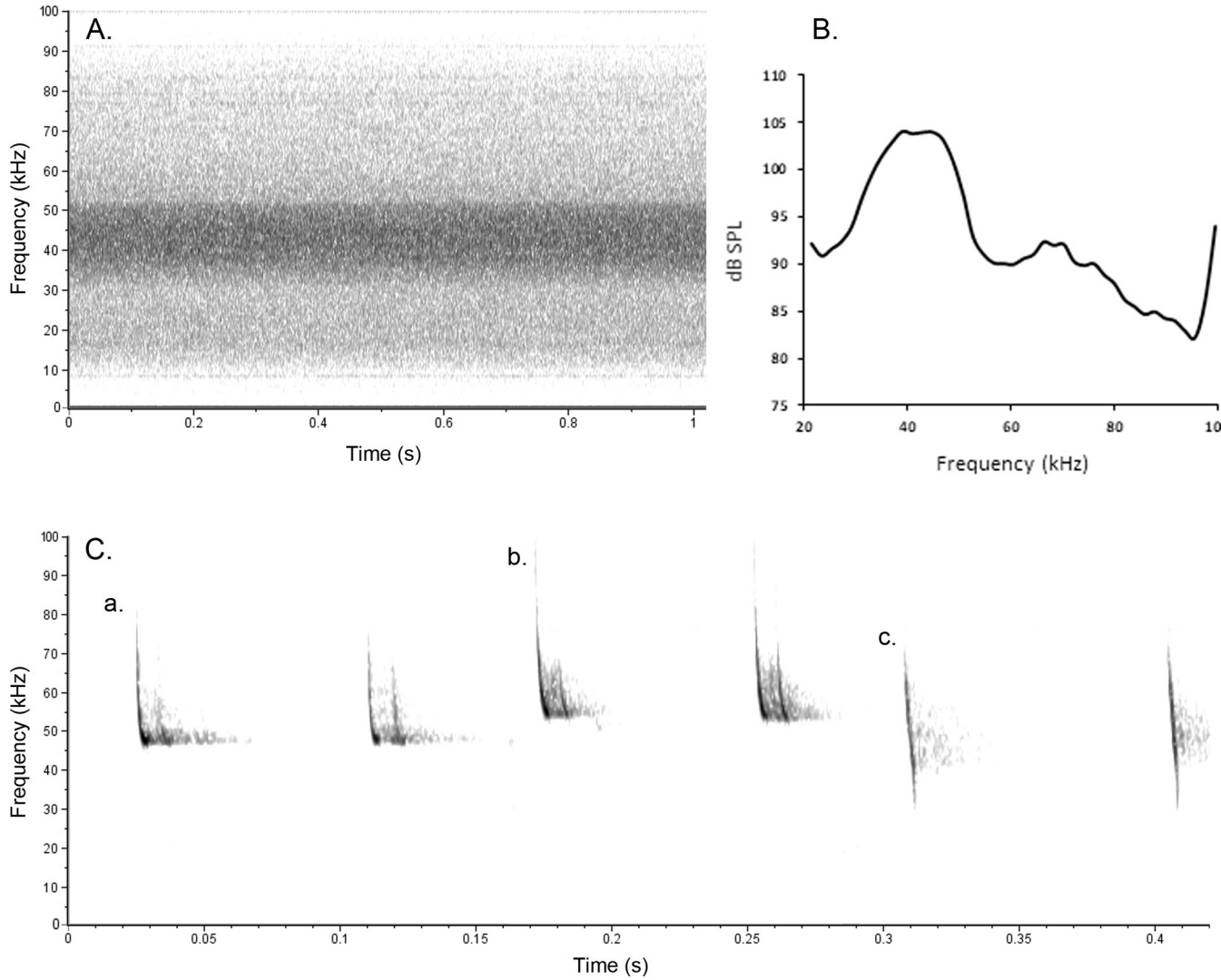

**Fig 2. Spectrogram and power spectrum of acoustic deterrent speaker output and spectrograms of echolocation calls of bat species recorded.**
Spectrogram (A.) and power spectrum (B.) calculated from acoustic deterrent speaker unit output recorded on-axis in an anechoic chamber using a Sanken CO-100K Super Wide Range Microphone and calibrated using a type 4231 Brüel & Kjær calibrator (114 dB SPL at 1 kHz). Power spectrum dB SPL measurements calculated at 1 m and adjusted for spectral sensitivity of microphone and distance (original distance 2.2 m, FFT 128, Hamming window, calculated in Avisoft SASLab Pro). Spectrograms of bat echolocation calls (C.), including two calls from a sequence identified as *Pipistrellus pipistrellus* (a.), *P. pygmaeus* (b.) and *Myotis* species (c.). Both spectrograms were created using Raven Lite 2.0.0 with the same settings (spectrogram window Hann, size 512, brightness 69, contrast 71).

buzzes in 10 s (a new pass was identified as a sequence of calls >1 s from the last and feeding buzzes are discernible due to their high repetition rate) [58, 59].

Speaker noise completely masked bat calls recorded at eight sites (peak frequency threshold of >40 kHz). For the remaining six sites, calls were discernible in ultrasound treatment files, but still less so than in radar and control files, where no deterrent sound was broadcast. We therefore added the deterrent noise to files recorded at six sites recorded during non-ultrasound treatments using MATLAB 2016a *sum wave files* function (Mathworks, Natick, USA), to avoid any bias introduced by some files being easier to score.

## Variables and model specification

We analysed the following response variables using GLMMs: visual bat activity (NIR video), acoustic bat activity (*Pipistrellus pipistrellus*, *P. pygmaeus* and *Myotis* species) and bat feeding buzzes for all species. We included the fixed effects of deterrent treatment (with levels: *radar*, *ultrasound*, *ultrasound and radar*, *control*), time block (levels: *A-D*), temperature and an inter-action term between deterrent treatment and time block in full models for the video data. We also included an interaction term between deterrent treatment and temperature, to examine any effects of temperature on the transmission of radar and/or ultrasound through the treatment area. However, temperature data were missing at one site, due to equipment failure and were substituted for the mean of all sites (<10% of data). Full models for acoustic data were the same but did not contain temperature data, due to missing data representing >10%. We also did not include an interaction term between deterrent treatment and time block in acoustic models, due to the low sample size precluding the use of a more complex model (rank deficiency). We retained the random effects of site (*N* = 14 for NIR data, *N* = 6 for acoustic data) and time block (*A-D*) in final models to control for the repeated measures design and to reduce pseudoreplication.

## Statistical analysis

We analysed bat pass count data with generalized linear mixed effect models (GLMMs), with a Poisson family and log link function in R (v3.6.1), using the lme4 package (v.1.1–21)[60]. We followed the backwards step-wise model selection method to find the most parsimonious, yet best-fit model for the data (final model) [57, 61]. We removed terms sequentially from a more complicated model when likelihood ratio tests (LRT) were non-significant and retained signif-icant terms in the final model. LRT statistics are presented as $\chi^2$, df and $p$ value and were obtained from LRTs between a model containing a term, and a restricted model without that term (or the null model). We present effect sizes and standard errors for final models in a table and post-hoc Tukey contrast test results in the text as $z$-statistics and $p$ values obtained using the multcomp package (v1.4–1) [62]. We validated final models by simulation using the R package DHARMa v.0.2.0, using residual plots to check for overdispersion, heteroscedasticity and zero inflation [63].

## Ethics statement

This study was approved by the University of Bristol Animal Welfare and Ethical Review Body (licence no: UB/17/045) and carried out in accordance with, and under strict recommenda-tions from the government licensing departments Natural England (2015-12272-SCI-SCI) and Natural Resources Wales (66141:OTH:CSAB:2015). Privately owned sites and those managed by the Canals and Rivers Trust were accessed with permission for all field experiments.

# Results

## Near-infrared video passes

We recorded a total of 3668 bat passes from 14 hours of near infrared (NIR) video footage (electronic supplementary material S1 Table) and a mean number of bat passes (± SD) of 489.07 ± 253.04 per site. The final model for NIR video data did not show overdispersion, zero-inflation or heteroscedasticity, when data were simulated using the R package DHARMa (v.0.2.0) [63]. The model included the fixed effect deterrent treatment, but there was no effect of temperature, time block or either interaction terms (between deterrent treatment and tem-perature/time block) (Table 1). Bat activity (NIR video passes) was significantly reduced when

an ultrasonic deterrent was deployed, but radar had no effect (Table 1 and Fig 3). Post-hoc Tukey contrast tests showed there was a significant reduction in bat activity, during ultrasound treatment blocks compared to the control periods and to radar treatment alone (81 and 84% reduction respectively) (ultrasound-control: $z = -4.63$, $p < 0.001$; ultrasound-radar: $z = -4.89$, $p < 0.001$). Bat passes were also significantly reduced (by 78 and 82%) when ultrasound and radar where deployed together, compared to the control, and to radar, respectively (Tukey contrasts: ultrasound & radar-control: $z = -4.30$, $p < 0.001$; ultrasound & radar-radar: $z = -4.56$, $p < 0.001$). Tukey contrasts showed there was no difference between ultrasound treatments and there was no reduction in bat activity when radar was compared to the control (ultrasound & radar-ultrasound: $z = -1.46$, $p = 0.46$; radar-control $z = 0.28$, $p = 0.99$).

## Acoustic passes

A total of 3073 acoustic bat passes were identified at 6 sites and a mean number of bat passes per site of 512.17 (± SD 287.79), including 1518 *Pipistrellus pygmaeus*, 618 *P. pipistrellus* and 388 *Myotis* species passes and 514 feeding buzzes (representing 49, 20, 13 and 17% of passes) (electronic supplementary material S2–S5 Tables). *Rhinolophus hipposideros*, unknown *Pipistrellus* species, *Eptesicus serotinus* and *Nyctalus* species made up the remaining passes (1%). Final models for acoustic data were not overdispersed or zero-inflated and also did not show heteroscedasticity in DHARMa (v.0.2.0) simulations [63]. Final models for *P. pipistrellus* data included the fixed effect of deterrent treatment, but there was no effect of time block on the number of passes recorded. *P. pipistrellus* passes were significantly reduced by 52 and 79% respectively during ultrasound treatments (*ultrasound* only; *ultrasound and radar*, compared to *control*). There was also a 39 and 72% reduction in *P. pipistrellus* passes during the *ultrasound* only, and *ultrasound and radar* treatments respectively, when compared to the *radar* only treatment. However, only *ultrasound and radar* treatments were significantly different in Tukey tests (Table 1 and Fig 3; Tukey contrasts: ultrasound & radar-control: $z = -3.75$, $p < 0.01$; ultrasound & radar-radar: $z = -2.61$, $p < 0.05$; ultrasound-control: $z = -2.34$, $p = 0.09$; ultrasound-radar: $z = -1.17$, $p = 0.65$; radar-control: $z = -1.18$, $p = 0.64$, ultrasound & radar-ultrasound: $z = -1.46$, $p = 0.46$). We recorded a 61 and 40% reduction in *P. pygmaeus* activity compared to the *control*, and a 56 and 33% reduction compared to *radar*, when *ultrasound* was deployed alone and combined with *radar*. However, the null model was the most parsimonious when *P. pygmaeus* data were analysed using a GLMM and deterrent treatment was marginally non-significant in LRT tests (Table 1). Despite this, *ultrasound* and *radar* were significantly different in Tukey contrast tests (ultrasound-radar: $z = -2.63$, $p < 0.05$; ultrasound-control: $z = -2.01$, $p = 0.93$; ultrasound & radar-control: $z = -1.44$, $p = 0.47$; ultrasound & radar-radar: $z = -2.05$, $p = 0.17$; radar-control: $z = 0.61$, $p = 0.93$; ultrasound & radar-ultrasound: $z = 0.56$, $p = 0.94$). *Myotis* species activity was not affected by any of the deterrent treatments, with similar numbers of passes recorded for all treatments. Bat feeding activity was significantly reduced (by 79 and 69%) during the *ultrasound* treatment, compared to the *control* and *radar* treatments and by 48% during *ultrasound and radar* treatment when compared to the control. However, bat activity was higher (25% increase) in *ultrasound and radar* treatments when compared to *radar*, although no significant effect of the *ultrasound and radar* treatment, or *radar* was observed in Tukey contrast tests (ultrasound-control: $z = -3.57$, $p < 0.01$; ultrasound-radar: $z = -2.65$, $p < 0.05$; ultrasound & radar-control: $z = -1.87$, $p = 0.24$; ultrasound & radar-radar: $z = -0.92$, $p = 0.79$; radar-control: $z = -0.96$, $p = 0.77$; ultrasound & radar-ultrasound: $z = 1.75$, $p = 0.30$).

**Table 1. Final model estimates (± SE) and likelihood ratio test statistics for near infrared (NIR) video and acoustic bat pass data, including *Pipistrellus pipistrellus*, *P. pygmaeus* and *Myotis* spp. passes and feeding buzzes of all species.**

| Model | Model terms | Estimates | SE | $\chi^2$ | df | *p* |
|---|---|---|---|---|---|---|
| NIR video | (Intercept) | 3.32 | 0.28 | | | |
| | *Deterrent treatment* | | | 29.92 | 3 | < 0.001 |
| | Radar | 0.09 | 0.31 | | | |
| | Ultrasound | -1.51 | 0.33 | | | |
| | Ultrasound & radar | -1.39 | 0.32 | | | |
| | Time block | | | 0.69 | 3 | 0.88 |
| | Temperature | | | 1.53 | 1 | 0.22 |
| | Deterrent treatment * temperature | | | 3.63 | 3 | 0.30 |
| | Deterrent treatment * time block | | | 13.87 | 9 | 0.13 |
| | Random effects | Variance | SD | % total | | |
| | Time block (within site) (*N* = 4) | 0.62 | 0.79 | 59.12 | | |
| | Site (*N* = 14) | 0.43 | 0.65 | 40.88 | | |
| *P. pipistrellus* | (Intercept) | 3.62 | 0.36 | | | |
| | *Deterrent treatment* | | | 11.72 | 3 | < 0.01 |
| | Radar | -0.52 | 0.44 | | | |
| | Ultrasound | -1.06 | 0.45 | | | |
| | Ultrasound & radar | -1.76 | 0.47 | | | |
| | Time block | | | 0.04 | 1 | 0.84 |
| | Random effects | Variance | SD | % total | | |
| | Time block (nested within site) (*N* = 4) | 0.55 | 0.74 | 74.45 | | |
| | Site (*N* = 6) | 0.19 | 0.43 | 25.55 | | |
| Feeding buzzes | (Intercept) | 3.04 | 0.46 | | | |
| | *Deterrent treatment* | | | 10.8 | 3 | < 0.01 |
| | Radar | -0.36 | 0.38 | | | |
| | Ultrasound | -1.45 | 0.41 | | | |
| | Ultrasound & radar | -0.72 | 0.39 | | | |
| | Time block | | | 2.05 | 3 | 0.56 |
| | Random effects | Variance | SD | % total | | |
| | Time block (nested within site) (*N* = 4) | 0.35 | 0.59 | 29.2 | | |
| | Site (*N* = 6) | 0.84 | 0.92 | 70.8 | | |
| *P. pygmaeus* | Deterrent treatment | | | 7.36 | 3 | 0.06 |
| | Time block | | | <0.01 | 1 | 0.95 |
| | Random effects | Variance | SD | % total | | |
| | Time block (nested within site) (N = 4) | 0.84 | 0.92 | 64.12 | | |
| | Site (*N* = 6) | 0.47 | 0.69 | 35.88 | | |
| *Myotis* spp. | Deterrent treatment | | | 2.68 | 3 | 0.44 |
| | Time block | | | 3.78 | 3 | 0.29 |
| | Random effects | Variance | SD | % total | | |
| | Time block (nested within site) (N = 4) | 1.22 | 1.1 | 59.22 | | |
| | Site (*N* = 6) | 0.84 | 0.92 | 40.78 | | |

## Discussion

In this study, we have shown that radar was ineffective as a deterrent for bats at foraging sites, but acoustic methods showed considerable promise. Previous work suggested that bat activity was reduced at foraging sites in Scotland, UK, by radar deployment over a 10–30 m range [18]. However, we could not confirm this deterrent effect, using video nor acoustic data, despite

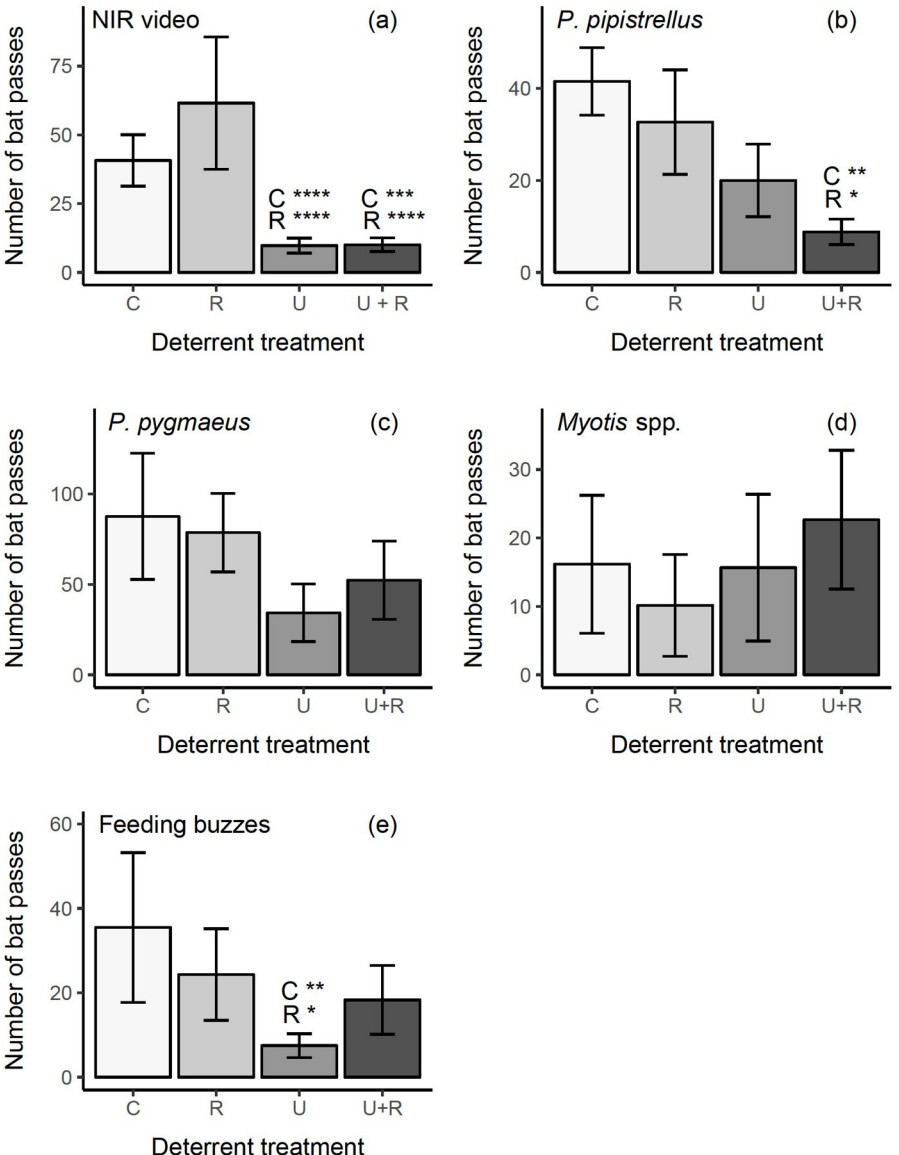

**Fig 3. Number of near infrared video (NIR) and acoustic bat passes recorded during control and deterrent treatments.** Bat activity (mean number of passes) during 4 treatments of ultrasound (U), radar (R), ultrasound and radar (U+R) and a control (C), including counts of bat passes recorded on near-infrared (NIR) video footage (a) and acoustic bat passes identified as *Pipistrellus pipistrellus* (b), *P. pygmaeus* (c), *Myotis* spp. (d) and feeding buzzes of all species (e). Including SE error bars and *p* values from post-hoc Tukey comparisons, presented as significance stars and associated label of treatment comparison (* = $p < 0.05$, ** = $p < 0.01$, *** = $p < 0.001$, **** = $p < 0.0001$). NIR video $N = 14$, acoustic data $N = 6$.

using similar methodology and recording over the same range of up to 30 m. The radar beam area projected by the unit used in both our study and the previous work in Scotland was highly directional. The horizontal and vertical beam angle of 1.9˚ and 22˚, respectively, would result in a treated airspace of ~0.7–1 m wide and 8–12 m tall (at ~20–30 m from the deterrents). In comparison, the 15˚ beam angle of the speakers would have resulted in a ~5–8 m treated airspace both horizontally and vertically. Therefore, bats may have only encountered the radar beam briefly at the centre of the treatment area, when flying through it, possibly limiting any potential for deterrence, if any exists. However, the speakers were positioned with the same line of site of the radar beam, with one speaker either side of the centre of the antenna (Fig 1)

and so were likely to have covered the same treatment area vertically. Despite this and the fact that the radar unit and methodology were very similar, it is unclear why the results of our study did not corroborate the findings from the Scottish study [18].

Bats are known to avoid ambient and broadcast noise and this study supports previous research in this area [12, 16, 45, 47, 48]. However, this study is the first to show a reduction in bat activity at foraging sites using broadcast ultrasound in a European context. We recorded a clear effect of ultrasound on bat activity with an ~80% reduction seen in the video footage data (when compared to control and radar treatments) (Fig 3). However, the acoustic data does not show such a clear trend, possibly due to unavoidable noise in the data from bat echolocation calls being recorded outside the range of the deterrent and small sample sizes that were unavoidable due to the deterrent masking echolocation calls at some sites. It was also not possible to discern feeding buzzes to species level due to high levels of overall activity and often more than one species in each file. Despite this, ultrasound treatments had the lowest mean number of bat passes recorded for all datasets, excluding *Myotis* species (Fig 3). We also found an effect of the ultrasound and radar treatment (when compared to control and radar only) and the ultrasound only treatment (when compared to radar) on *Pipistrellus pipistrellus* and *P. pygmaeus* activity respectively. There was also an effect of ultrasound only on feeding activity of all species. *Myotis* species were not significantly affected by any deterrent treatment.

The lack of response to acoustic deterrence by *Myotis* bats compared to *Pipistrellus* species, may have been due differences in hearing and/or the mechanisms underpinning deterrence. For example, the deterrent sound could have acted as an aversive stimulus rather than a masking one, as was the case in traffic noise playback experiments on *M. daubentonii* [46]. How a bat species responds to a deterrent sound may be in part due to its hearing threshold, which in bats typically range from about 20–30 dB, but may be as low as 0 dB [64–66]. Hearing thresholds of bats are largely understudied, but can depend on frequency, the hearing system of the individual and even the time of day [64–66]. Therefore, a difference in hearing thresholds may account for the differences in response to acoustic deterrence in this study. The different response to the deterrent of *Myotis* bats may also have been due to the experimental site selection and set-up. Due to the nature of the riparian sites, the *Myotis* species bats recorded were likely to be *Myotis daubentonii*, which feed directing their echolocation calls towards the water [67]. The *Myotis* bats may therefore have moved along the water during ultrasound treatments, away from the speakers, to nearer the bridge, where we had placed the bat detector, resulting in more passes recorded during ultrasound treatments. As *Pipistrellus* species are not limited to feeding over water, they may have dispersed in different directions when the sound was broadcast, rather than in the direction of the bat detector microphone. *Myotis nattereri* were deterred from specific roosting areas, with no habituation effect found after 15 days when the same acoustic deterrents as used in this study were tested in churches in Norfolk in the UK [12]. Therefore, a deterrent effect of ultrasound can not be ruled out for *Myotis daubentonii*, without a follow-up exploration of acoustic deterrents focussed on this species in this context (e.g. with speakers pointing at the water's surface).

Our findings support the premise that ultrasonic speakers show promise for use as bat deterrents. We therefore recommend that these methods be considered as a way of reducing impacts on bats from human structures and activities, and in situations where conservation conflicts arise from bats roosting in buildings. For example, these methods could be used at wind turbines to reduce bat fatalities in Europe, as has been done in North America [16]. Acoustic deterrence could also be used to manage bats that roost in buildings where people live, work, study and worship, alleviating cleaning burdens, damage and risks to human health. For example, the US Centre for Disease Control and Prevention (CDC) regards bats roosting in buildings as a reservoir for histoplasmosis and therefore recommends exclusion to prevent

the threat to human health [68]. However, currently the CDC guidelines explicitly state that deterrents are ineffective in this context. Therefore, we recommend updating these guidelines to reflect the potential of ultrasonic deterrence.

Another application for acoustic deterrence is as part of transport infrastructure mitigation. For example, acoustic deterrence methods could be used in combination with existing mitigation methods (e.g. over/underpasses, green bridges), to prevent bats from crossing roads or train-lines, where they may be at risk from collision and mortality [28]. For example, acoustic deterrence has been tested as a potential mitigation strategy for use in areas where woodland bats may be affected by new train lines [69]. Current bat mitigation strategies for roads are at present regarded as largely insufficient [28]. Methods such as overpasses that are currently ineffective [70], could therefore be used in combination with acoustic deterrence, allowing bats to utilise safer flight lines over roads.

Despite its potential, acoustic bat deterrence may have welfare implications that need to be considered when weighing up its use against alternative mitigation methods. Deterrence, by its design, acts to move animals out of an area, which may mean a reduction in access to foraging resources or habitat features (such as roosts or commuting routes). If these resources are rare in the remaining habitat not affected by deterrence, or if the noise degrades surrounding habitats due to overspill effects, this could lead to unwanted impacts on the target species (and non-target species). However, small-scale movement of bats to new roosts within churches using acoustic deterrence did not affect where radio-tracked bats from the same church went to forage and also had no other obvious impacts on behaviour [12, 39]. Bats are likely to respond to an acoustic deterrent stimulus in a similar way to that from an anthropogenic source, such as traffic or gas compressor noise [46–49]. Anthropogenic noise can induce measurable stress responses in some animals such as an increase in cortisol levels or heart rate [32]. However, any results of stress indication should also be weighed up against the short and long-term effects of the human activity being mitigated for, which may cause more harm to bats in the long term.

In conclusion, acoustic deterrence shows great potential for use as a mitigation measure, to reduce human impacts on bats in a wide range of contexts. However, we found no evidence of radar being an effective deterrent. We caution against the wide-spread use of acoustic deterrence as a universal solution to bat mitigation and recommend exploration of its use on a case-by-case basis, following the mitigation hierarchy [31]. Displacement of bats may often seem a legitimate way forward for conservation, but as with any mitigation measure, the methods need to be weighed up against less invasive methods, avoiding any potentially detrimental effects. Further work therefore needs to focus on safe, effective, practical and targeted implementation of deterrents in the applications discussed. As species-specific responses to acoustic deterrence may also influence the efficacy and safe use of these methods, behavioural mechanisms underpinning deterrence should also be explored, along with differences in hearing and the distance over which these methods are effective for different species.

## Supporting information

**S1 Table. Number of bat passes recorded on near infrared video during deterrent treatments and control.**
(DOCX)

**S2 Table. Number of *Pipistrellus pygmaeus* passes recorded during deterrent treatments and control.**
(DOCX)

**S3 Table. Number of *Pipistrellus pipistrellus* passes recorded during deterrent treatments and control.**
(DOCX)

**S4 Table. Number of *Myotis* spp. passes recorded during deterrent treatments and control.**
(DOCX)

**S5 Table. Number of feeding buzzes recorded during deterrent treatments and control.**
(DOCX)

## Acknowledgments

We thank the Bats and Wind Energy Cooperative for use of their speakers. We also thank all the project volunteers for their hard work, with special thanks to Eleanor Field for carrying out video analysis. We thank all the landowners of the sites used in the study for their cooperation, including Canals and Rivers Trust and private landowners that do not wish to be named.

## Author Contributions

**Conceptualization:** Lia R. V. Gilmour, Gareth Jones.

**Data curation:** Lia R. V. Gilmour.

**Formal analysis:** Lia R. V. Gilmour.

**Funding acquisition:** Simon P. C. Pickering, Gareth Jones.

**Investigation:** Lia R. V. Gilmour, Marc W. Holderied, Simon P. C. Pickering, Gareth Jones.

**Methodology:** Lia R. V. Gilmour, Marc W. Holderied, Simon P. C. Pickering, Gareth Jones.

**Project administration:** Lia R. V. Gilmour, Gareth Jones.

**Resources:** Lia R. V. Gilmour, Marc W. Holderied, Gareth Jones.

**Software:** Marc W. Holderied.

**Supervision:** Marc W. Holderied, Simon P. C. Pickering, Gareth Jones.

**Validation:** Lia R. V. Gilmour, Marc W. Holderied, Simon P. C. Pickering, Gareth Jones.

**Visualization:** Simon P. C. Pickering.

**Writing – original draft:** Lia R. V. Gilmour.

**Writing – review & editing:** Lia R. V. Gilmour, Marc W. Holderied, Simon P. C. Pickering, Gareth Jones.

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
