## [Decision Letter · Decision Letter 0]

27 Oct 2019

PONE-D-19-23958

Deterrence as a mitigation measure to reduce conflict between bats and humans

PLOS ONE

Dear Dr Gilmour,

Thank you for submitting your manuscript to PLOS ONE. After careful consideration, we feel that it has merit but does not fully meet PLOS ONE’s publication criteria as it currently stands. Therefore, we invite you to submit a revised version of the manuscript that addresses the points raised during the review process. In particular, Reviewer 2 has raised a number of concerns that should be addressed in full for your paper to be accepted for publication.

We would appreciate receiving your revised manuscript by December 6, 2019. To enhance the reproducibility of your results, we recommend that if applicable you deposit your laboratory protocols in protocols.io, where a protocol can be assigned its own identifier (DOI) such that it can be cited independently in the future. For instructions see: http://journals.plos.org/plosone/s/submission-guidelines#loc-laboratory-protocols

We look forward to receiving your revised manuscript.

Kind regards,

Michelle L. Baker, PhD

Academic Editor

PLOS ONE

Journal Requirements:

2. In your Methods section, please provide additional location information of the experimental sites, including geographic coordinates for the data set if available.

This study was funded by the National Environment Research Council (NERC) (NE/K007610/1), with CASE contribution from Ecotricity (www.ecotricity.co.uk), received by LRVG and PI was GJ. The funders had no role in study design, data collection and analysis, decision to publish, or preparation of the manuscript.

We note that you received funding from a commercial source: Ecotricity Group Limited

Reviewers' comments:

Reviewer's Responses to Questions

**Comments to the Author**

1. Is the manuscript technically sound, and do the data support the conclusions?

Reviewer #1: Yes

Reviewer #2: Partly

2. Has the statistical analysis been performed appropriately and rigorously? 

Reviewer #1: Yes

Reviewer #2: No

3. Have the authors made all data underlying the findings in their manuscript fully available?

Reviewer #1: Yes

Reviewer #2: Yes

4. Is the manuscript presented in an intelligible fashion and written in standard English?

Reviewer #1: Yes

Reviewer #2: Yes

5. Review Comments to the Author

Reviewer #1: This manuscript is very well written and describes a straightforward set of experiments testing the effects of broadband noise and radar on localized bat activity. There are only a few previous studies on the topic, and none were nearly as rigorous and thorough as this one. The experiments provide what I consider to be the best evidence so far that broadband noise can be used to deter bats from using or entering an area. The study also does a good job of challenging the hypothesis that radar can be an effective deterrent, which was dubious from the start but very difficult to test. The large sample size and the rigorous comparison of the differences in results obtained with audio and video also makes this paper very strong. Thus, I think this paper will become highly cited in the field and a cornerstone of future efforts.

I only have a few suggestions and comments.

1) The manuscript provides compelling evidence of species differences in the response to the ultrasonic deterrent. This species effect is worthy of being mentioned in the abstract even if the underlying reasons are not yet clear, because this should be a point of emphasis for future studies.

2) I would recommend including the supplementary figure S1 in the main text, and I suggest adding either representative spectrograms or power spectra of the bat pulse emissions to illustrate the extent to which the stimulus overlapped with the main bat species being studied. Alternatively, in the methods section after line 91, it would be helpful to provide the reader with a brief description of the pulse acoustic properties in relation to the stimulus acoustic properties. The extent of overlap between the bandwidth of the noise and the pulses may be important for predicting species differences in the response.

3) Line 91: for completeness, do you have information about the beam projection angle? The radar beam angle was provided on line 86, and it was discussed that the narrow beam angle may be the reason radar was ineffective. However, the ultrasonic beam angle is needed for comparison.

4) Introduction, and Line 63-64. An important motivation for deterring bats from buildings that is not mentioned in the introduction/discussion is that bat colonies pose health hazards, particularly when they invade schools or other public buildings. The US Center for Disease Control (CDC) guidelines for dealing with bats explicitly states that ultrasonic deterrents are ineffective (https://www.cdc.gov/niosh/docs/2005-109/pdfs/2005-109.pdf ), citing Tuttle, 1988. This topic might be something worth addressing in the manuscript.

Typos: line 78 (latest/lasted)

Line 230 (moved/moving?)

Reviewer #2: Manuscript Number: PONE-D-19-23958

Manuscript Title: Deterrence as a mitigation measure to reduce conflict between bats and humans

This study aimed at testing two acoustic measures to deter bats from their foraging sites: ultrasonic sound and radar, alone and in combination. While acoustics play an important role in the lives of bats, the rationale for doing this study somewhat missed me. As the authors indicate, there already exists evidence of bats being deterred from wind turbines (and churches). Thus this study only corroborated this without adding more knowledge, e.g. regarding why bats are deterred. I also have some concerns regarding the framing of the manuscript. They focus is on deterrence in general, while the literature cited mainly is from wind turbine deterrence. While deterrence may be an option for mitigation, when other measures (e.g. avoid, minimize) have been tried I fail to see how it can be employed in the general sense. The authors mention, besides wind turbines, also deterring bats from roads and foraging sites. However, both would have detrimental effects to bat populations; the latter depriving them from their food sources and the first creating barrier effects. I guess this was unattended by the authors, but the framing of using acoustics indiscriminately does neither help bats nor humans in my eyes. Below follow more specific comments:

L11: “impacts of threats” – what is this, did the authors mean “risk of impact”?

L12: “sensitive areas such as protected artifacts” is somewhat unclear to me: sensitive to whom, bats of humans? Why not just say historic buildings or human structures?

L12-13: specify where you deterred bats from, i.e. their riparian foraging sites.

L16: Pp echolocation call activity was reduced by ~40-80% not 70-80%! Only the combination of ultrasound + radar was significant. This needs to be clarified. Include also the non-significant effects on Myotis and P pygmeus.

L17-19: Why would you suggest that only ultrasound should be used as deterrent, when in some of the cases only the combination significantly reduced bat activity?!

L22: I find the wording of “parties” somewhat misleading: are bats a party (politically, socially or other) in this? What do you mean by “impact”; mortality of individuals, loss of habitat or population reductions?

L23: I guess that in 99% of the time it is wildlife that is impacted by humans and not vice versa. This, either by taking into use species’ habitats, or killing them when they cause nuisance.

L25-26: maybe better to say that we are mainly talking about human-human conflicts here?!

L28: I would suggest using “human structures and activities” (omitting “-built”) as then “human” links to both structures and activities; now it doesn’t.

L31: “…where they may come to harm” or cause harm to human structures and activities (i.e., avoiding conflict).

L33-34: Be clearer why bats would need to be deterred. All references are from wind turbines apart from one; would it not be more realistic to either extend the scope of structures/activities or set specific focus on wind energy?

L34-35: “…bats are already…” already vulnerable relative to what? One example given (habitat loss) is mainly human-caused; isn’t that part of the human-wildlife conflict? Or do you see HWC mainly as bats causing nuisance for humans?

L39-42: In this example I fail to clearly see the stakeholder conflict. What is this conflict centered on?

L42-43: Maybe indicate how bats may be affected by roads. Also I fail to see the link to deterrence here.

L47: Before going into reducing impacts, may clarify that in all cases the mitigation hierarchy should be the leading line (avoid, minimize, reduce, compensate, restore). Also, clarify when deterrence may be implemented: in itself it may enhance functional habitat loss and barrier effects for bats, so maybe not preferred over other more benign measures?! The references that are supposed to cover light, sound and radar seem to miss those relating to light (e.g. [23,24]), and what about [12]?

L52: I would suggest moving this sentence after L52-55.

L57: Bats avoid insect noise? If this was actually meant, please include references (also for turbulent water).

L58-60: All these references relate to anthropogenic noise? Still the statement is generic, and should therefore also include natural causes of noise.

L61: What is the difference between communication and eavesdropping exactly? Is eavesdropping not a specific form of communication? Also, I would suggest explaining what eavesdropping is for the non-expert.

L66: Why was light not also included? While it is mentioned above, this is not included. Explain why the focus lies on acoustics.

L65: Can you give examples where bats actually cause conflict? Where would deterrence be needed, contrary to other mitigation measures?

L77: Isn’t this called “time blocks” in the models? Please clarify.

L78: Did the experiments last (spelling mistake here) 1 hour or 55 minutes? (4x 10 min + 3x 5 min in-between = 55 minutes)

L80: Was this alternation only done spatially or also temporarily?

L84: 0.3 milli sec (ms) or 0.3 micro sec (μs)?

L87-89: how directional were the speakers exactly? Was their sound emission measured? Could it also be that the narrow horizontal radar beam caused non-significant effects in this measure contrary to the ultrasound? What would the results be if the radar was tilted 90 degrees giving a narrow vertical beam but wide horizontal beam? Also, I would suggest inclusion of a picture presenting the exact practical layout of the design spatially.

L90: What was the peak power of the speakers?

L97: 20m from the radar or speakers or camera? Unclear.

L114: “in” instead of “during”?

L115-117: I don’t understand, the results speak of 6 sites being included in the analyses, here it states that all were included after having added deterrent noise. Which is it?

L118-132: I would move this section after the next “Variables and model specification”. In addition, it seems you have used both a information theoretic AND frequentist approach in the statistical model/term selection. This is somewhat confusing, I would urge you to choose one of the two.

L126: “restricted model” instead of “nested model”?

L127: How were the final model validated exactly? I do not find any information on this in the results.

L130-132: I do not follow this sentence.

L133-143: What was the generic design of the study, did it follow a Control-Impact design? If so, maybe it is more illustrative to only show the full models to be able to see the effects or non-effects of the treatment?! Unless you wish to better understand in what circumstances deterrence worked well (or not); such as temperature or wind speed. I was somewhat baffled to notice that wind speed, which clearly affects both bat activity and sound emission and ambient noise, was not included in the models. Was ambient background noise measures at all?

L153: maybe clarify that this is the mean number of bat passes and give the unit (i.e. per day, per experiment, per hour, per 10-minute interval, per minute?).

L156: “per time unit” – what is the time unit used? Be clear.

L166 (table): What is the difference between the fixed effect “Time block” and the random effect “Block”? Why was an interaction between deterrent and temperature included? Should also an interaction between deterrent and time block be considered?

L181-185: No results are given regarding which of the models were most parsimonious (Table S2). This table clearly indicates that the most parsimonious model (the one with lowest AIC, whereby usually ΔAIC<2 are seen as being equally good) would be the null model and not the deterrent treatment model?! This would mean that none of the echolocation models rendered any significant outcomes.

L196-197: Were related Tukey tests also non-significant?

L203-204: What were the results for all bat species pooled regarding acoustic passes? This would enable a direct comparison to the visual video-footage.

L205 & L214: This is not a chapter I mam afraid…

L215-216: Tja, although done at a new site, this study merely corroborates earlier finding. What new insights have been obtained?

L222: Discuss why the combination of ultrasound and radar led to a better result.

L228-230: Why? The radar beam has a wide vertical beam which should be expected to cover both the water surface and the airspace above (unless it was placed too close to the foraging site). I have not seen any information regarding the directionality of the speakers (horizontally and vertically). Is there also any specifics of the riparian sites regarding habitat (e.g. treed vegetation, topography) that may have affected the results?

L230: the bats “may have been moved”? By whom? Maybe rephrase.

L239: Are there clear difference in hearing between these species that might explain their responses (or lack thereof)?

L240-241: Why would anyone wish to exclude bats from their foraging sites, especially riparian areas? What may be expected to be built there (not likely to be wind turbines or churches)? What human activities will conflict with bat presence?

L244-246: Are you really suggesting to deter bats from roads, increasing barrier effects roads already pose even more? I fail to see how this may help conserve bat species. Reference [22] suggests underpasses instead. Although roads may affect bats through noise, your study fails to give any indication how bats are affected by ambient noise.

L246: What other human structure and activities, other than wind turbines and roads, are you thinking of here?

L247-248: What “specific scenarios” are you thinking of here? Different spatial planning and development scenarios? Scenarios of locating different structures or activities and their expected effect? It is unclear to me what is meant here.

L249: The distance over which speakers or radar have an effect in deterring bats should preferentially have been part of this study.

L249-250: This comes back to my earlier question whether the different results might have been caused by differences in hearing among species? The same rationale would also matter for radar (e.g. different emission bands)? Please add references to indicate to which extent this would be possible.

L251.255: This is in fact the first time you indicate that possible deterrence may not be the first and foremost preferred measure to implement. PI think this point should be raised already in the introduction, together with an indication when it might be best to implement (and when not). Now the manuscript seems to say that acoustic deterrence will work ate wind turbines, churches, roads, foraging sites, etc. Also, something that may be discussed is how practical the implementation of acoustic deterrence is: can it really be implemented along (long stretches of) roads?

Cheers Roel May

6. PLOS authors have the option to publish the peer review history of their article (what does this mean?). If published, this will include your full peer review and any attached files.

Reviewer #1: Yes: Michael Smotherman

Reviewer #2: Yes: Roel May

---

## [Author Response · Author response to Decision Letter 0]

4 Dec 2019

School of Biological Sciences

University of Bristol 

24 Tyndall Avenue

Bristol BS8 1TQ

03/12/19

Rebuttal letter for re-submission to PLoS ONE

Manuscript reference: PONE-D-19-23958

Manuscript title: Comparing acoustic and radar deterrence methods as mitigation measures to reduce human-bat impacts and conservation conflicts

Dear Dr Baker, 

Thank you for your invitation to submit a revised version of our manuscript for consideration by PLoS ONE. The submitted comments from both reviewers were very helpful in allowing us to provide a revised manuscript that thoroughly introduces the topic and background of bat deterrence methods, as well as framing our research within the field. 

Reviewer 1’s (R#1) comments are very encouraging and complimentary and they clearly understood the main aims of the study and how it contributes to the field of bat deterrence methods. R#1 also describes our methods of using acoustic and near infrared video as a rigorous and our approach thorough. We welcome their suggestions to include bat echolocation call spectrograms for comparison with the deterrent output and thank them for drawing our attention to the US CDC guidelines. 

R#2 provided comments and questions in the detailed line-by-line analysis of the manuscript that were very helpful in allowing us to identify areas of the manuscript that needed more information or description. Many of the comments and criticisms given by R#2 were likely due to the manuscript being too concise and have been easily be remedied by additional information. We hope that the rationale behind the study is now clearer.

We also acknowledge that R#1’s background is in bat ecology, bioacoustics and conservation, unlike R#2. Therefore, the context in which this study sits and the wider issues it addresses may be less obvious for someone with a background in broader areas of conservation, such as R#2. It is therefore useful to have both reviewers contrasting comments, as it allows us to make sure we frame the manuscript, so that it is explicit in what the study is trying to achieve, the issues it addresses and its applicability in different scenarios. The comments have therefore been very helpful in allowing us to make sure the manuscript is useful for a wider readership, not just those who have a specific interest in bat conservation. 

Below is a general and line-by-line response to each of the referee’s comments. We have also made the required changes (where possible) to the manuscript in line with your request to adhere to the journal requirements (see below). We hope that our careful attention to the reviewers’ comments will now make our manuscript suitable for publication in PLoS ONE.

Yours sincerely, 

Lia Gilmour

- We have checked all our file names and made sure the manuscript meets PLOS ONE’s style requirements. 

- In your Methods section, please provide additional location information of the experimental sites, including geographic coordinates for the data set if available.

- We have included a range of geographic locations for the areas that we carried out experiments in on L136-137. 

- However, we were asked by the landowners of our sites not include the exact locations of these sites and therefore cannot provide locations to any higher level of resolution.

- Thank you for stating the following in the Financial Disclosure section…Please provide an amended Competing Interests Statement that explicitly states this commercial funder, along with any other relevant declarations relating to employment, consultancy, patents, products in development, marketed products, etc

- As requested, we have provided an amended competing interests statement and financial disclosure statement (below), explicitly stating the commercial funder and the contribution provided towards the PhD stipend of Lia Gilmour, the student carrying out this PhD project. 

Financial Disclosure Statement

This study was funded by the National Environment Research Council (NERC) (NE/K007610/1), with CASE contribution from Ecotricity (www.ecotricity.co.uk), received by LRVG and PI was GJ. The funders (NERC) had no role in study design, data collection and analysis, decision to publish, or preparation of the manuscript. SPCP of Ecotricity Group Limited also provided a supervisory role in the project and review of the manuscript before submission for publication.

Competing Interests Statement

This study was part of a PhD funded by the National Environment Research Council (NERC) (NE/K007610/1). As part of the PhD funding, £3000 was provided towards stipend costs for LRVG as a CASE contribution by the commercial funder Ecotricity Group Limited (www.ecotricity.co.uk). SPCP of Ecotricity Group Limited also provided a supervisory role in the project and review of the manuscript before submission for publication. This does not alter our adherence to PLOS ONE policies on sharing data and materials.

Rebuttal letter for re-submission to PLoS ONE

Manuscript reference: PONE-D-19-23958

Manuscript title: Comparing acoustic and radar deterrence methods as mitigation measures to reduce human-bat impacts and conservation conflicts

General response to Reviewer #1:

R#1: 

“This manuscript is very well written and describes a straightforward set of experiments testing the effects of broadband noise and radar on localized bat activity. There are only a few previous studies on the topic, and none were nearly as rigorous and thorough as this one. The experiments provide what I consider to be the best evidence so far that broadband noise can be used to deter bats from using or entering an area. The study also does a good job of challenging the hypothesis that radar can be an effective deterrent, which was dubious from the start but very difficult to test. The large sample size and the rigorous comparison of the differences in results obtained with audio and video also makes this paper very strong. Thus, I think this paper will become highly cited in the field and a cornerstone of future efforts.”

Response:

- We thank R#1 for such positive comments about our manuscript and experimental design. We believe that they have understood the rationale for the study and the need to test and compare the two bat deterrence methods.

- We also thank R#1 for drawing attention to our “rigorous comparison” of results from both video and acoustic data, which stands our study out from the small number previously done.

Line-by-line response to R#1 comments:

I only have a few suggestions and comments.

1) The manuscript provides compelling evidence of species differences in the response to the ultrasonic deterrent. This species effect is worthy of being mentioned in the abstract even if the underlying reasons are not yet clear, because this should be a point of emphasis for future studies.

- We agree that it is important to include the differing results in terms of species and have therefore added in a more detailed explanation of the results in the abstract, including a line pointing out the differences between the species. L23-29.

- These comments are also in line with some of the comments from R#2. We accept that we may have not been clear with the overall message and outcome of the results regarding the NIR video data and how the acoustics back this up. We have therefore made this clearer in the abstract and included the key results from models and % reduction figures. We believe this makes the message clearer and allows the reader to get a more detail understanding of the main results with regards to the overall trend (NIR video data) and species information (acoustic data).

- In response to R#2’s comment regarding L196-197, we also reanalysed the data and carried out Tukey contrast tests for Pipistrellus pygmaeus pass count data. Although there was no significant effect of deterrent treatment overall on P. pygmaeus, there was a significant effect of ultrasound when compared to radar for this species. Therefore, we have updated the results and abstract accordingly and stated that we see a deterrent effect of ultrasound on both Pipistrellus species, but not Myotis species. 

2) I would recommend including the supplementary figure S1 in the main text, and I suggest adding either representative spectrograms or power spectra of the bat pulse emissions to illustrate the extent to which the stimulus overlapped with the main bat species being studied. Alternatively, in the methods section after line 91, it would be helpful to provide the reader with a brief description of the pulse acoustic properties in relation to the stimulus acoustic properties. The extent of overlap between the bandwidth of the noise and the pulses may be important for predicting species differences in the response.

- In line with R#2’s comments we have created a new figure including a spectrogram of the three species’ echolocation calls incorporated into what was Figure S1. This new figure should allow comparison of species’ calls with the acoustic deterrent output and illustrate the extent of overlapping frequencies, as suggested.

- We also make a point of explaining that the deterrent’s output overlaps with the species’ echolocation calls in the text (L167-170). We have also given the frequency of most energy for each species/species group in the text to draw the reader’s attention to how the “stimulus acoustic properties” relate to the bat species’ echolocation call properties.

3) Line 91: for completeness, do you have information about the beam projection angle? The radar beam angle was provided on line 86, and it was discussed that the narrow beam angle may be the reason radar was ineffective. However, the ultrasonic beam angle is needed for comparison.

- We have included the beam angle for the speaker unit transducers (L167). 

4) Introduction, and Line 63-64. An important motivation for deterring bats from buildings that is not mentioned in the introduction/discussion is that bat colonies pose health hazards, particularly when they invade schools or other public buildings. The US Center for Disease Control (CDC) guidelines for dealing with bats explicitly states that ultrasonic deterrents are ineffective (https://www.cdc.gov/niosh/docs/2005-109/pdfs/2005-109.pdf ), citing Tuttle, 1988. This topic might be something worth addressing in the manuscript.

- We thank R#1 for drawing our attention to the US CDC guidelines and also to the potential wider application for bat deterrence in buildings in general; where bat roost (i.e. not just in churches, which we had already referred to). 

- In response to these comments by R#1 and some comments by R#2 regarding the specific applications for bat deterrence, we have made changes to the abstract, introduction and discussion. We have introduced the context of bats in historic buildings and also when bats roost in other buildings such as workplaces, schools etc. We have also included a reference to the health risks to people in buildings where bats roost. 

- We have also included a recommendation to update the US CDC guidelines for dealing with bats that roost in buildings, so that they include acoustic deterrence as a potential method for exclusion (L374-378). 

General response to Referee #2:

Reviewer #2: 

Manuscript Number: PONE-D-19-23958

Manuscript Title: Deterrence as a mitigation measure to reduce conflict between bats and humans

This study aimed at testing two acoustic measures to deter bats from their foraging sites: ultrasonic sound and radar, alone and in combination. While acoustics play an important role in the lives of bats, the rationale for doing this study somewhat missed me. As the authors indicate, there already exists evidence of bats being deterred from wind turbines (and churches). Thus this study only corroborated this without adding more knowledge, e.g. regarding why bats are deterred. I also have some concerns regarding the framing of the manuscript. They focus is on deterrence in general, while the literature cited mainly is from wind turbine deterrence. While deterrence may be an option for mitigation, when other measures (e.g. avoid, minimize) have been tried I fail to see how it can be employed in the general sense. The authors mention, besides wind turbines, also deterring bats from roads and foraging sites. However, both would have detrimental effects to bat populations; the latter depriving them from their food sources and the first creating barrier effects. I guess this was unattended by the authors, but the framing of using acoustics indiscriminately does neither help bats nor humans in my eyes. 

Response:

The comments from R#2 are thorough and helpful, although we believe that they may have misunderstood our rationale and intentions with regards to the study. We acknowledge that the manuscript was over concise and may have therefore seemed to generalise our results to applications that do not seem relevant. However, we believe that having elaborated on the rationale and application scenarios, we have alleviated many of the issues raised in the review. Most of the comments have been easily remedied with additional information, which we have provided in response to the line-by-line comments below. 

However, we do not accept that this study simply corroborates previous studies. It is the first study to directly compare two methods for deterrence in bats and also to do so in the European context at foraging sites. Previous acoustic deterrence research has been carried out at wind turbines in North America and in roosts in historic buildings in the UK, which both represent different contexts and species. Radar has been tested as a deterrent in foraging sites in the UK but has not been compared directly to other types of deterrence. As stated by R#1, the findings of the studies examining radar as a deterrent carried out at UK bat foraging sites are considered controversial. Our study therefore aimed to replicate the studies testing radar as a potential deterrent and also compare the methods with acoustic deterrence in the same context. Our results do not corroborate the findings of those previous studies testing radar at foraging sites and therefore we believe our findings are novel and important, as explained by R#1. R#2 also seems to misunderstand that we are testing acoustic methods and a form of electromagnetic radiation, which is not acoustic. While the suggested mechanism for radar’s effect on animals may have some acoustic component, this has not been tested. Therefore, it is not clear why R#2 is placing so much emphasis on our study being solely about acoustics, when radar is also not explicitly regarded as an acoustic method of deterrence. 

Regarding R#2’s comments on the framing of the manuscript, we accept that the Introduction does not elaborate enough on the potential applications of deterrence and focusses only on the wind farm scenario. Therefore, we have ameliorated this by including a more detailed introduction to acoustic deterrence applications (L52-71). We have also added more information on how these methods could be used in the discussion, as well as welfare implications and suggestions for future research (L368-410). For example, R#2 also had misgivings about the use of deterrence in the context of roads and other areas where bats forage, but where they may be at risk from human activities. We have therefore updated the manuscript with a caveat that acoustic deterrence be used on a case-by-case basis and discuss some of the potential impacts of using deterrence. We also discuss that deterrence may be an important tool in conserving bat populations in areas where they are at risk of impacts or indeed mortality (such as at wind turbines or roads) and that there may be a trade-off with the associated impacts of the acoustic deterrence.

In the line-by-line comments, there are a number of points where R#2 picks up on issues with our use of the phrase ‘human-wildlife conflict’ or similar. We acknowledge that our use of the phrase was somewhat confusing and although many papers in the field still use it, we have made substantial changes to the manuscript removing it. Instead, we have included an introduction to the alternative phrases ‘human-wildlife impacts’ and ‘conservation conflicts’, in line with suggestions in the literature. We have also changed the title to include these phrases. We believe the manuscript now provides a clear introduction to these phrases before using them in relation to bat deterrence methods and applications. We thank R#2 for picking up on this as it has allowed us to be more accurate in describing how our work fits in the wider fields of human wildlife impacts and conservation conflicts. 

Line-by-line response to R#2:

Below follow more specific comments:

L11: “impacts of threats” – what is this, did the authors mean “risk of impact”?

- We have reworded this to make the sentence clearer, to “impacts of human structures or activities on wildlife” L12-13

L12: “sensitive areas such as protected artifacts” is somewhat unclear to me: sensitive to whom, bats of humans? Why not just say historic buildings or human structures?

- We agree that this was an unclear phrase and was alluding to historic artefacts in churches and protecting these from urine and faeces. 

- We have reworded the whole sentence to make the point clearer and to incorporate all human built structures and activities and buildings where bats may be regarded as a nuisance or health hazard. We have also made these changes after the advice from R#1 (point 4), who suggested we mention public buildings more generally and not just historic buildings such as churches. L12-14

L12-13: specify where you deterred bats from, i.e. their riparian foraging sites.

- We have updated this sentence to make it clear that we are testing two methods that have shown potential in the literature for deterring bats in a number of scenarios, including where bats may forage (i.e. around roads, wind turbines) or roost (in buildings, bridges). L15

L16: Pp echolocation call activity was reduced by ~40-80% not 70-80%! Only the combination of ultrasound + radar was significant. This needs to be clarified. Include also the non-significant effects on Myotis and P pygmeus.

- We have updated the abstract to include all results for the different species and have altered the % reduction for P. pipistrellus to ~40-80%. 

- We quoted the 70-80% reduction initially as this was the range of % reduction seen in the ultrasound and radar (U+R) treatment as compared to the control and radar only. However, we understand this may have been misleading. 

- All species’ % reductions have been updated in the same way, giving the whole range of all ultrasound treatments (U and U+R) as compared to both control and radar only treatments. L23-25

L17-19: Why would you suggest that only ultrasound should be used as deterrent, when in some of the cases only the combination significantly reduced bat activity?!

- The results from the near infrared video footage show a clear effect of both ultrasound treatments, but no effect of radar. 

- It is also clear from Fig 3 that for both the Pipistrellus species and feeding buzzes, the trend is a reduction in activity in both ultrasound treatments and in all cases the radar treatment has a higher mean number of bat passes. 

- We would therefore deduce that it is the ultrasound that is having an effect and radar is not. 

- The species information is a little less clear, probably due to a smaller sample size, unavoidable due to the masking effect of the deterrent making it impossible to identify bat passes at some sites for the acoustic data. 

- However, we do see a significant effect of ultrasound only treatment in post-hoc tests (in the case of feeding buzzes, when compared to both radar and control treatments, and P. pygmaeus, when compared to radar) and also in combination with radar in P. pipistrellus (when compared to both control and radar).

- But we do not see an effect of radar on its own in any of the tests. Indeed, bat activity in ultrasound and radar treatments is significantly reduced when compared to radar and radar is therefore unlikely to be having an effect on bat activity. 

- We have added a sentence to the abstract making the lack of an effect of radar in either video or acoustic data clearer L19-20

L22: I find the wording of “parties” somewhat misleading: are bats a party (politically, socially or other) in this? What do you mean by “impact”; mortality of individuals, loss of habitat or population reductions?

- We have reworded this, taking out the word “parties” as suggested by R#2.

- We have also added in examples of impacts so this statement is more clear. L34-35

L23: I guess that in 99% of the time it is wildlife that is impacted by humans and not vice versa. This, either by taking into use species’ habitats, or killing them when they cause nuisance.

- We agree with this point and have updated the sentence to reflect this (please see the previous point). 

L25-26: maybe better to say that we are mainly talking about human-human conflicts here?!

- We agree that the phrase ‘human-wildlife conflict’ (HWC) is a misleading one and we have therefore changed the manuscript to include the alternative terms ‘human-wildlife impacts’ and ‘conservation conflicts’ (terms suggested by Redpath et al. 2015 references [2-3]).

- We have reworded the title to include these terms instead of HWC, as well as the rest of the manuscript (abstract, introduction and discussion)

- We have also included an explanation of why the phrase HWC is misleading in lines… of the introduction and introduce the alternative phrases of ‘human-wildlife impacts’ and ‘conservation conflicts’. L36-40 

L28: I would suggest using “human structures and activities” (omitting “-built”) as then “human” links to both structures and activities; now it doesn’t.

- We have done as R#2 suggests and removed “-built” from this line and also in all places we have used it in the manuscript. 

L31: “…where they may come to harm” or cause harm to human structures and activities (i.e., avoiding conflict).

- We have reworded this section so that it includes the human element, which is suggested in the phrase above by R#2. L44-46

L33-34: Be clearer why bats would need to be deterred. All references are from wind turbines apart from one; would it not be more realistic to either extend the scope of structures/activities or set specific focus on wind energy?

- The reason the references are mainly from wind turbines is that there has not been that much published on bat deterrence in any other contexts to date.

- Although we can not add in more references for different applications, we have included more explicit reference to the other potential applications and scenarios where bat deterrence could be used. L47-71.

L34-35: “…bats are already…” already vulnerable relative to what? One example given (habitat loss) is mainly human-caused; isn’t that part of the human-wildlife conflict? Or do you see HWC mainly as bats causing nuisance for humans?

- We have removed the phrase “bats are already” and any mention of human-wildlife conflict (please see L25-26 response).

- We have now rephrased these lines explaining the threats generally to bats, irrespective of the human context. L50-52

- This is part of a general re-wording of the whole section, where we have added in more detail and removed any reference to human-wildlife conflict and instead referring to ‘human-wildlife impacts’ and/or ‘conservation conflicts’.

L39-42: In this example I fail to clearly see the stakeholder conflict. What is this conflict centered on?

- We have removed this example as it does not clearly explain the message we are trying to convey and have replaced it with some background about the impact of wind turbines on bats in the US and European context. L53-59

L42-43: Maybe indicate how bats may be affected by roads. Also I fail to see the link to deterrence here.

- As suggested, we have added in some examples of impacts of roads on bat populations (habitat loss, noise pollution and mortality due to collision).

- In this section we are introducing potential applications for deterrence, where bats are at risk from human structures and/or activities.

- We have added in two sentences, firstly after the wind turbine application introduction and secondly after the section on roads/transport infrastructure. We explain in these sentences how deterrence might be used to reduce impacts on bats, or more specifically to reduce direct mortality due to collision with either turbine blades or vehicles. L58-64

L47: Before going into reducing impacts, may clarify that in all cases the mitigation hierarchy should be the leading line (avoid, minimize, reduce, compensate, restore). Also, clarify when deterrence may be implemented: in itself it may enhance functional habitat loss and barrier effects for bats, so maybe not preferred over other more benign measures?! The references that are supposed to cover light, sound and radar seem to miss those relating to light (e.g. [23,24]), and what about [12]?

- We agree that bat conservation should be the focus of any mitigation measure that aims to reduce impacts of human structures/activities on bats.

- We welcome the suggestion to include reference to the mitigation hierarchy and have added in a paragraph that relates to bats. L72-80

- We have also added in an example of possible unwanted side effects of deterrents on target and non-target animals, as suggested by R#2. 

- Our aim is to make sure the reader fully understands where bat deterrence is a viable method for reducing human impacts on bats and conservation conflicts involving bats. But we do not wish to advocate its widespread use for any bat related issue. We therefore welcome R#2’s comments and believe that they enhance and clarify our message. 

L52: I would suggest moving this sentence after L52-55.

- We have done as R#2 suggests and moved the sentence explaining the studies on the effect of radar on bats after the sentence about the suggested mechanism for deterrence. L101-107

- We have also added more detail about the two Scottish studies involving radar, to allow more accurate and informed comparison of our study in the discussion.

L57: Bats avoid insect noise? If this was actually meant, please include references (also for turbulent water).

- Bats did indeed avoid areas where high frequency insect noise was broadcast in a study on Brazilian free-tailed bats in the USA. However, we have added in the missing reference Gillam et al. 2007 (reference number [44]) and thank R#2 for picking this up. L109-111

- The reference relating to turbulent water was already included as reference, Mackey & Barclay, 1989 [45], but we have inserted it along with reference [44] after the first mention and included the references relating to anthropogenic noise pollution [46-49] at the end of the sentence, for increased clarity.

L58-60: All these references relate to anthropogenic noise? Still the statement is generic, and should therefore also include natural causes of noise.

- We have included reference [44] and [45] which relate to insect noise and noise from turbulent water. L111-113

L61: What is the difference between communication and eavesdropping exactly? Is eavesdropping not a specific form of communication? Also, I would suggest explaining what eavesdropping is for the non-expert.

- We have reworded the reference to eavesdropping to explain its difference to communication and why it may be beneficial to a bat. Bats tend to ‘eavesdrop’ on the echolocation calls of other bats, which are produced when bats are foraging or orientating in their environment. Eavesdropping is therefore a passive form of listening in, rather than active communication. L113-116 

L66: Why was light not also included? While it is mentioned above, this is not included. Explain why the focus lies on acoustics.

- The focus of this study is on acoustics and radar, not just acoustics, which are differing methods that have shown potential as deterrents for bats. Radar is a type of electromagnetic radiation, which differs from sound waves. 

- The mechanism for the effect of radar (if there is one) is unknown. It is suggested by the authors of the Scottish radar studies [18-19] that there may be an acoustic component, in that the brain of bats may vibrate, producing a noise, which causes the deterrent effect, but this has not been tested. We therefore regard the two methods as separate. 

- Light was not included due to its potentially serious detrimental effects on roosting and foraging bats.

- We realise that we did not introduce light deterrents in enough detail, which has been picked up by R#2. Therefore, we have added in an explanation of why they were left out of the study and have included a section detailing the effect of light deterrents and light in general on bats and given examples of why light deterrents should, in our opinion, not be used. L81-100.

L65: Can you give examples where bats actually cause conflict? Where would deterrence be needed, contrary to other mitigation measures?

- We have reworded this sentence in line with the manuscript-wide changes to remove the phrase ‘human-wildlife conflict’, please see response to point about L25-26. We have instead referred to ‘impacts’ and ‘conservation conflicts’. L124-126.

L77: Isn’t this called “time blocks” in the models? Please clarify.

- We have clarified by changing “treatment blocks” to “ treatment time blocks”. L140-141.

L78: Did the experiments last (spelling mistake here) 1 hour or 55 minutes? (4x 10 min + 3x 5 min in-between = 55 minutes)

- The experiments lasted for one hour and each treatment time block was followed by a 5-min recovery period. 

- We have updated the sentence to reflect this. L140-141.

L80: Was this alternation only done spatially or also temporarily?

- This alternation was only done temporally following a temporal Latin square design, initially deployed over 16 sites, although due to equipment failure at two sites, we could only use 14 in analysis. 

- However, generalized linear mixed effect models allow for unbalanced design. 

- Please see response to point about line L133-143 below for a more detailed explanation of the experimental design and changes we have made to the manuscript for increased clarity with regards to statistics performed. 

- The topography of the riparian sites and the way in which bats used them prevented us from following a spatial design. Bats occupied areas nearer bridges present at the site earlier on, when light levels were high and flew more over the water away from the bridges as light levels decreased over the experiment hour. Therefore, deploying the deterrents in different areas spatially within the site was not technically feasible and would have reduced our ability to record an effect of either deterrent method. 

L84: 0.3 milli sec (ms) or 0.3 micro sec (μs)?

- We have changed it the mistake to 0.3 μs. L158-159.

L87-89: how directional were the speakers exactly? Was their sound emission measured? Could it also be that the narrow horizontal radar beam caused non-significant effects in this measure contrary to the ultrasound? What would the results be if the radar was tilted 90 degrees giving a narrow vertical beam but wide horizontal beam? Also, I would suggest inclusion of a picture presenting the exact practical layout of the design spatially.

- The point about the speakers’ beam projection angle was also picked up by R#1 point 3. 

- We have included the beam projection angle of 15° at -6 dB, provided by Senscomp specification from www.Senscomp.com in L167.

- The aim of our experiment was to compare radar and acoustic methods in a similar way as had been done in previous studies. We wanted to replicate the methods and equipment used in the Scottish radar study in order to hopefully corroborate their findings, however we did not find an effect of radar. We followed Nicholls & Racey’s methodology and deployed a similar marine radar unit, using the same specifications, in a similar way at riparian foraging sites. As they were able to find an effect of radar on bat activity in their study, this seemed an effective way to compare it to acoustic methods. 

- It is possible that the narrow horizontal radar beam caused non-significant effects in our study, but as it was very similar equipment and methodology as used in the Scottish radar study, it is unclear why we did not find the same result. 

- We have calculated the area of treated airspace from the beam angles for both the radar and speaker units, to enable more accurate comparison of the deterrent propagation in the environment. We have included this as part of our Discussion. L317-329

- We have created a schematic of the experiment set-up and included it in the methods section, as well as referring to it in the discussion. L152, L326

L90: What was the peak power of the speakers?

- We do not have information on the peak power of the speaker units/transducers, but have provided information of the source dB level at 1 m and at 20 m, for the 50 kHz (the frequency of maximum energy). L162-167

- In response to R#1’s comments we have also included Fig S1 in the main text, including a spectrogram and a power spectrum of the deterrent output, as well as spectrograms of relevant bat species echolocation calls. L170

L97: 20m from the radar or speakers or camera? Unclear.

- The SM2 bat detector was placed at ~20 m from the cameras, which were at 0 m of the treatment zone, in line with the camera. We have updated this sentence to include a reference to the camera. L155

- We have included a schematic detailing where all the equipment was placed and set-up at a site, which we refer to in the methods and also the discussion. L152, L326

L114: “in” instead of “during”?

- Changed L206

L115-117: I don’t understand, the results speak of 6 sites being included in the analyses, here it states that all were included after having added deterrent noise. Which is it?

- Only recordings from six sites were included in analysis as bat calls were completely masked by deterrent noise in all other sites. 

- We have added “files recorded at six sites” to the second sentence, to make it clear that ultrasound was added to only those remaining six sites i.e. those that were not completely masked by ultrasound. L205-210

L118-132: I would move this section after the next “Variables and model specification”. 

- We agree, it makes more sense to move this section and have done as R#2 suggests. L211-226

In addition, it seems you have used both a information theoretic AND frequentist approach in the statistical model/term selection. This is somewhat confusing, I would urge you to choose one of the two.

- We acknowledge that our approach to modelling and presentation of model selection statistics was confusing. Our method for model selection was a frequentist approach using likelihood ratio tests (LRTs), including p values obtained from comparing models containing a term with one with that term removed. We also obtained AICc values and pseudo R squared values to compare the models, which are generally more used in the information theoretic (IT) approach. 

- Therefore, as suggested by R#1 we have now focussed on the frequentist approach only and have made the following changes to the manuscript: 

- We have removed the model selection statistics tables presented in Table S2 and focussed on the frequentist approach only (as AICc values are usually used in the information theoretic approach rather than the frequentist approach)

- We have re-run the models using only the frequentist approach and checked the statistics against the tables originally submitted. 

- Re-running the models produced no change to the statistics from those originally presented. 

- However, there was a mistake in the original manuscript regarding the GLMM distribution, which was Poisson, rather than negative binomial. L227-239. 

L126: “restricted model” instead of “nested model”?

- We have changed “nested model” to “restricted model” as R#2 suggests, as the latter is a clearer phrase describing a model with a term removed. L234 

L127: How were the final model validated exactly? I do not find any information on this in the results.

- We validated models using the R package DHARMa, which uses simulation of the data to produce simulated residual plots and tests for overdispersion, zero inflation and heteroscedasticity. Use of this package for validation is becoming more widespread, as it allows simulation of the data and validation of Poisson and negative binomial count data, which are often difficult to validate using normal residual plots. See https://cran.r-project.org/web/packages/DHARMa/vignettes/DHARMa.html for more information.

- We have updated the methods to make the method of validation clearer. L237-239

- We have included a sentence the both the NIR video and acoustic data results sections explaining that final models did not show overdispersion, zero-inflation or heteroscedasticity when tested with the R package DHARMa and we have included the reference again. We have also included which models were chosen as the final models in the results section. L250-252, L284-285.

L130-132: I do not follow this sentence.

- We have reworded the statistical analysis section and removed this sentence as it was confusing. The explanation of our methods is now clearer and explains how LRT statistics were obtained. L227-239.

L133-143: What was the generic design of the study, did it follow a Control-Impact design? 

If so, maybe it is more illustrative to only show the full models to be able to see the effects or non-effects of the treatment?! 

Unless you wish to better understand in what circumstances deterrence worked well (or not); such as temperature or wind speed. I was somewhat baffled to notice that wind speed, which clearly affects both bat activity and sound emission and ambient noise, was not included in the models. Was ambient background noise measures at all?

- We followed a temporal Latin square design, alternating treatment order (four treatments) over 16 sites. However, we had an equipment failure which meant we couldn’t use two of the sites in the analysis. We therefore used GLMMs which control for unbalanced designs and also included time block as a fixed effect in the models. 

- These methods follow a within-subject or repeated measures design, where each treatment is tested within one subject, or in our case each site. We chose to use these methods as they allow analysis of effects on bat activity, which can vary significantly between sites and on different nights. Therefore, testing all four treatments within one night, but alternating those treatments on different nights was, in our minds, the best way to test these deterrent methods on bats. 

- It is normal practice to present effect sizes for final models rather than full models and also LRT statistics for terms/fixed effects included in those models, when using backwards stepwise model selection methods. 

- We did not measure ambient background noise, as any differences between sites were controlled for using the within-site design. Our aim was simply to test the different deterrent methods at sites where bat activity was high enough to record differences (i.e. which is why we chose riparian sites). Recording ambient noise would therefore not have added to our results or conclusions in this instance. 

- We did not include wind speed or rainfall in the models and did not measure these variables during experiments as we purposely chose nights where there was no rainfall and wind speeds were below 10 m/s. Wind speed and rainfall can significantly reduce bat activity. Carrying out experiments in higher wind speeds would have resulted in reduced numbers of bat passes being recorded, making it more difficult to see a difference in the effects of the treatments. Therefore, in order to get meaningful results and test these deterrents effectively on bats, we needed to ensure sufficient bat activity as baseline at chosen sites. This is not to say that these environmental variables are not important to consider in deterrent application. Future studies should indeed include an investigation into the effects of weather variables and other factors such as site topography on the spread of high frequency sound in the environment. 

- We have therefore updated the methods in light of R#2’s comments, explaining our study design in more detail and making both the experimental design and statistics methods clearer. L143-149 

- We have also added in our rationale behind not including wind speed and rainfall in our study design. L172-173 

L153: maybe clarify that this is the mean number of bat passes and give the unit (i.e. per day, per experiment, per hour, per 10-minute interval, per minute?).

- The mean and SD we give was calculated as a mean per site (for 14 sites) and we have updated the results to reflect this as suggested. L249-250

- We have done the same for the acoustic bat passes results section. L280

L156: “per time unit” – what is the time unit used? Be clear.

- We agree this phrase is confusing and have removed it from the sentence, instead referring to ultrasound treatment blocks. L286-287

L166 (table): What is the difference between the fixed effect “Time block” and the random effect “Block”? Why was an interaction between deterrent and temperature included? Should also an interaction between deterrent and time block be considered?

- There is no difference between the fixed or random effect time block, they are the same variable. We have updated the Table to reflect this, with the same name fixed and random effect variable. L265-268

- We included an interaction between temperature and treatment as the propagation of sound and EM waves can be affected by the temperature of the air the waves are travelling through. This might therefore affect how bats respond to the deterrent.

- We did originally include an interaction between deterrent treatment and time block in the NIR video models and this was non-significant, so we have included it in the table. 

- However, due to low sample sizes, acoustic models that included an interaction between deterrent treatment and time block were rank deficient (i.e. not enough data to use a more complex model). Therefore, specifying a more complex model (i.e. with an interaction term) was more likely to cause inaccurate results. We therefore included the simpler models in analyses to avoid spurious results. 

- We have therefore updated the reasoning for inclusion (or not) of the two interaction terms in lines and also included the interaction term results in Table 1 for the NIR video response variable. L212-226 

L181-185: No results are given regarding which of the models were most parsimonious (Table S2). This table clearly indicates that the most parsimonious model (the one with lowest AIC, whereby usually ΔAIC<2 are seen as being equally good) would be the null model and not the deterrent treatment model?! This would mean that none of the echolocation models rendered any significant outcomes.

- Please see response to point made on L118-132 regarding approach to modelling. 

- We re-ran the models and found there was a mistake in Table S2 due to a comparison being made between a negative binomial model and the full model, rather than a Poisson model. 

- However, we have removed Table S2 as AIC values were associated with an information theoretic approach to modelling and we have now focussed on a frequentist approach. 

- We have included information on which models were the most parsimonious (and therefore chosen as the final models) for all response variables in the Results section and which also explained which effects were significant and which were non-significant. L252-254, L285-287

L196-197: Were related Tukey tests also non-significant?

- We have re-run the Tukey tests for the P. pygmaeus data and included the results in the manuscript. L299-303 

- We have also updated the discussion and abstract to reflect this result. 

L203-204: What were the results for all bat species pooled regarding acoustic passes? This would enable a direct comparison to the visual video-footage.

- We did not run models for all species’ acoustic data pooled as we did not think this would add anything more to the results and conclusions. 

- Species’ responses to the deterrent were different and therefore pooling all species and running models would likely give an outcome skewed towards one particular species and does not give us any more accurate information than looking at species individually.

- We also do not believe the pooled acoustic data would be comparable to the video data, as it was recorded at 20 m from the camera/deterrent set up, covering a range of ~30 m in all directions, to certain degrees depending on the species. Therefore, the data includes a lot of noise and is not necessarily representative of the treatment area filmed on the cameras.

- We included the species acoustic information in the manuscript to mainly get an idea of the species present during experiments that may have been deterred in the video data.

- It is very difficult to record accurate species data that is equivalent to video without placing the bat detectors in the same place as the video cameras, or indeed closer to the cameras than we did, which was not possible due to the masking effect of the deterrent sound on bat echolocation call data. Indeed, we could only use 6 out of 14 of the sites that acoustic data were recorded for, due to this problem of masking. 

- Despite this, we did find differences in species responses to the acoustic deterrent.

L205 & L214: This is not a chapter I mam afraid…

- Changed “chapter” to “study” L314

L215-216: Tja, although done at a new site, this study merely corroborates earlier finding. What new insights have been obtained?

- We do not accept that our study is merely corroborating earlier findings on bat responses to ultrasound. Please see the response to the general comments made by R#2 on this matter at the beginning of the review response. 

L222: Discuss why the combination of ultrasound and radar led to a better result.

- We do not believe that the combination of ultrasound and radar led to a ‘better’ result, it is more likely an artefact of a smaller sample size and noise introduced into the data by the range of SM2 bat detector microphone. 

- Please see more detailed response this point in response to comment about line L17-19

- We have added in a point about the overall trend of reduced bat activity during ultrasound treatments seen in Fig 3. L339-340 

- We have updated the Tukey results for P. pygmaeus and added a comment on this in the discussion. P. pygmaeus activity was reduced during ultrasound only treatments when compared to radar, but there was no effect on combined treatments. L340-344

L228-230: Why? The radar beam has a wide vertical beam which should be expected to cover both the water surface and the airspace above (unless it was placed too close to the foraging site). I have not seen any information regarding the directionality of the speakers (horizontally and vertically). Is there also any specifics of the riparian sites regarding habitat (e.g. treed vegetation, topography) that may have affected the results?

- We are referring here to the directionality of the acoustic deterrent system not the radar, however we agree with the point generally that the acoustic deterrent beam would be expected to cover the surface of the water and have taken this sentence out. 

- It is more likely that as Myotis bats feed over the water, they are more likely to have been deterred away along the river closer to the bat detector during deterrent treatments, compared to the Pipistrellus species which feed higher above the water. 

L230: the bats “may have been moved”? By whom? Maybe rephrase.

- We have rephrased this to “may have moved”. L357

L239: Are there clear difference in hearing between these species that might explain their responses (or lack thereof)?

- Hearing sensitivity and thresholds in bats are largely understudied but could play a part in different responses of bat species to acoustic deterrent sounds. 

- There also may be some differences in the mechanism underpinning deterrence and how the species feed over water, which may explain why an effect of acoustic deterrence was not seen in Myotis species. 

- We have updated the discussion to include reference to bat hearing thresholds and the potential mechanisms for acoustic deterrence. L348-353

L240-241: Why would anyone wish to exclude bats from their foraging sites, especially riparian areas? What may be expected to be built there (not likely to be wind turbines or churches)? What human activities will conflict with bat presence?

- The premise of this study was to examine bat responses to acoustic and radar deterrents as proof-of-concept for their use in other applications.

- The reason we chose riparian foraging sites, was due to the high bat activity at these sites, which enabled us to detect an effect of the deterrent methods more easily.

- We acknowledge that the phrasing of this sentence seems centred around foraging sites. Therefore, we have removed the reference to foraging sites in order to make the sentence more general. L367 

- We have also re-written the discussion, to include specific examples where acoustic deterrence may have application. We appreciate that as it was, the discussion seemed to suggest that deterrence was just intended to remove bats from foraging sites and this was no our aim. Therefore, we have added specific examples relating to wind farms, roads/train lines and buildings, where deterrence may be useful. L368-387

L244-246: Are you really suggesting to deter bats from roads, increasing barrier effects roads already pose even more? I fail to see how this may help conserve bat species. Reference [22] suggests underpasses instead. Although roads may affect bats through noise, your study fails to give any indication how bats are affected by ambient noise.

- Current mitigation for roads is deemed currently insufficient (see Altringham & Kerth 2016 [28]). Therefore, acoustic deterrence has potential in being used alongside current structures such as green bridges and overpasses, in order to divert bat flight lines over these structures or other safer alternative routes over roads. 

- We have updated the discussion and introduction to include a more detailed explanation of the transport infrastructure application. L379-387

- Acoustic deterrence also has potential for use in reducing impacts of train lines on bats and we have included a reference to a blog detailing some work we did in collaboration with industry partners, Wevill et al. 2019 [70]. L382-384

- We have also updated the discussion with a section on welfare implications, including the effects of ambient and anthropogenic noise on bats. L388-401

L246: What other human structure and activities, other than wind turbines and roads, are you thinking of here?

- We have taken this sentence out and instead focussed on specific examples of applications (please see responses to two previous points).

L247-248: What “specific scenarios” are you thinking of here? Different spatial planning and development scenarios? Scenarios of locating different structures or activities and their expected effect? It is unclear to me what is meant here.

- We have removed this phrase and included specific applications for deterrence more generally (please see previous points on roads etc). 

L249: The distance over which speakers or radar have an effect in deterring bats should preferentially have been part of this study.

- The Scottish radar study [19] found a deterrent effect of radar over 30 m. We replicated their methodology and failed to find an effect of radar over these distances. 

- We believe that investigating the effect of the acoustic deterrents is out of the remit of this study, which had the main aim of comparing two bat deterrent methods that have shown promise in the literature, but that have not been compared at foraging sites in the UK or Europe. 

- We have carried out experiments examining the response of bats to the acoustic deterrent speakers over different distances and are currently preparing a separate manuscript including these results.

L249-250: This comes back to my earlier question whether the different results might have been caused by differences in hearing among species? The same rationale would also matter for radar (e.g. different emission bands)? Please add references to indicate to which extent this would be possible.

- We have included a discussion about bat hearing thresholds in relation to differing species response to deterrence. Please see response to point about L239. 

- We have not discussed hearing in relation to radar as we did not find an effect of radar and would therefore not seem relevant to the discussion. 

- We have however, included a reference to bat hearing in the recommendations for future work. L409-414

L251.255: This is in fact the first time you indicate that possible deterrence may not be the first and foremost preferred measure to implement. PI think this point should be raised already in the introduction, together with an indication when it might be best to implement (and when not). Now the manuscript seems to say that acoustic deterrence will work ate wind turbines, churches, roads, foraging sites, etc. Also, something that may be discussed is how practical the implementation of acoustic deterrence is: can it really be implemented along (long stretches of) roads?

- We have included in both the introduction and discussion reference to the mitigation hierarchy in response to the point raised. L75, L406

- We have also made sure we are clear in explaining that deterrence should be considered once all other more benign or less invasive alternatives have been exhausted. L402-414

- We have included a more detailed and explicit discussion of the relevant applications of deterrents, being careful to point out that they should only be used on a case-by-case basis. We have also updated the introduction, so that we introduce potential scenarios where deterrence might be applicable. 

- We have also included a section on welfare implications and finish on a warning that deterrents should not be rolled out for use anywhere a bat is regarded a ‘nuisance’.

- We are not suggesting that deterrence be implemented along long stretches of roads, but rather at key points to divert key bat flight lines along safer routes and we have updated the discussion and introduction to reflect this. L379-387 

- We have also explained that when deterrence is used it should be done so in an effective, safe, practical and targeted manner. L409-410

---

## [Editor Report · Decision Letter 1]

22 Jan 2020

Comparing acoustic and radar deterrence methods as mitigation measures to reduce human-bat impacts and conservation conflicts

PONE-D-19-23958R1

Dear Dr. Gilmour,

We are pleased to inform you that your manuscript has been judged scientifically suitable for publication and will be formally accepted for publication once it complies with all outstanding technical requirements.

With kind regards,

Michael Smotherman

Academic Editor

PLOS ONE

Additional Editor Comments (optional): I apologize for the lengthy delay in getting this processed. Thank you for your carefully attention to the revisions.  I'm certain this paper will be of great interest to many in the field.
---

## [Editor Report · Acceptance letter]

28 Jan 2020

PONE-D-19-23958R1 

Comparing acoustic and radar deterrence methods as mitigation measures to reduce human-bat impacts and conservation conflicts 

Dear Dr. Gilmour:

I am pleased to inform you that your manuscript has been deemed suitable for publication in PLOS ONE. Congratulations! Your manuscript is now with our production department. 

With kind regards,

on behalf of

Dr. Michael Smotherman 

Academic Editor

PLOS ONE